# PANC-1 Cell Line as an Experimental Model for Characterizing PIVKA-II Production, Distribution, and Molecular Mechanisms Leading to Protein Release in PDAC

**DOI:** 10.3390/ijms25063498

**Published:** 2024-03-20

**Authors:** Antonella Farina, Sara Tartaglione, Adele Preziosi, Patrizia Mancini, Antonio Angeloni, Emanuela Anastasi

**Affiliations:** Department of Experimental Medicine, Sapienza University of Rome, Policlinico Umberto I, 00181 Rome, Italy; antoeffe22@gmail.com (A.F.); sara.tartaglione@uniroma1.it (S.T.); adelepreziosi@gmail.com (A.P.); patrizia.mancini@uniroma1.it (P.M.); antonio.angeloni@uniroma1.it (A.A.)

**Keywords:** PDAC, PIVKA-II, glucose, epithelial to mesenchymal transition

## Abstract

Pancreatic ductal adenocarcinoma (PDAC) represents a highly aggressive malignancy with a lack of reliable diagnostic biomarkers. Protein induced by vitamin K absence (PIVKA-II) is a protein increased in various cancers (particularly in hepatocellular carcinoma), and it has recently exhibited superior diagnostic performance in PDAC detection compared to other biomarkers. The aim of our research was to identify an in vitro model to study PIVKA-II production, distribution, and release in PDAC. We examined the presence of PIVKA-II protein in a panel of stabilized pancreatic cancer cell lines by Western blot analysis and indirect immunofluorescence (IFA). After quantitative evaluation of PIVKA-II in PaCa 44, H-Paf II, Capan-1, and PANC-1, we adopted the latter as a reference model. Subsequently, we analyzed the effect of glucose addiction on PIVKA-II production in a PANC-1 cell line in vitro; PIVKA-II production seems to be directly related to an increase in glucose concentration in the culture medium. Finally, we evaluated if PIVKA-II released in the presence of increasing doses of glucose is concomitant with the expression of two well-acknowledged epithelial–mesenchymal transition (EMT) markers (Vimentin and Snail). According to our experimental model, we can speculate that PIVKA-II release by PANC-1 cells is glucose-dependent and occurs jointly with EMT activation.

## 1. Introduction

Ninety percent of pancreatic cancer cases are attributed to pancreatic ductal adenocarcinoma (PDAC), characterized by its aggressive and lethal nature [1]. With a 5-year survival rate of approximately 8%, PDAC stands as the sole cancer exhibiting an escalating mortality rate for both men and women. Nowadays, surgery remains the most radical treatment option for PDAC, but only 15–20% of cases are deemed surgically resectable at the time of diagnosis [2]. PDAC is in fact commonly referred to as the “silent killer” owing to its characteristic late diagnosis, with only 7% of cases identified at an early stage due to the absence of specific early symptoms; unfortunately, the tumor becomes apparent only after infiltrating surrounding tissues or metastasizing [3]. Given the tight correlation between survival rates of PDAC patients and disease stage, an early detection of the neoplasm becomes a pressing necessity to significantly enhance treatment effectiveness [4]. To date, there are no screening or surveillance programs for the early diagnosis of PDAC. The primary tool for determining the localization, extent, and clinical staging of the mass relies on imaging techniques. In the last few decades, there has been a significant increase in emphasis on circulating biomarkers as early warning systems for assessing disease risk. They have become a potent and cost-effective tool in cancer management [5]. However, in the case of PDAC, compared to other solid neoplasms, the clinical utilization and subsequent benefits of biomarkers remain considerably limited. Current guidelines regarding PDAC suggest Carbohydrate Antigen 19.9 (CA19.9) as the gold-standard circulating biomarker. However, CA19.9 has several limitations, since altered levels have been observed in patients with non-cancerous conditions such as biliary obstructions, chronic pancreatitis, and non-malignant jaundice. Additionally, not all PDAC patients exhibit elevated CA19.9 levels, particularly in Lewis-negative individuals, meaning its diagnostic accuracy is significantly reduced [6]. The suboptimal diagnostic performance of CA19.9 and the absence of other molecules indicating the presence of PDAC underscore the compelling need to explore new biomarkers with enhanced sensitivity and specificity. In recent years, there has been growing interest in the protein induced by vitamin K absence (PIVKA-II), an immature form of prothrombin also known as DCP (Des-gamma-Carboxy Prothrombin). Prothrombin, a vitamin K-dependent coagulation factor, is naturally synthesized by the liver under physiological conditions. In instances of vitamin K deficiency or when its action is hindered, such as with the administration of antivitamin-K drugs, PIVKA-II is released into the bloodstream [7].

Several lines of evidence underscore the connection between the absence of vitamin K and cancer. Vitamin K, an essential nutrient, has recently been explored as a potential anticancer agent, as demonstrated by its ability to inhibit the survival of certain pancreatic cancer cell lines through apoptotic mechanisms [8,9]. The deficiency of vitamin K can be detected using molecules such as PIVKA-II. This aspect has garnered significant interest within the scientific community [6].

Although PIVKA-II has proven to be a recognized tool in the diagnosis and prognosis of hepatocellular carcinoma (HCC) [10,11,12], it has also been observed that serum PIVKA-II alone could be a reliable biomarker for the detection of pancreatic cancer, showing superior diagnostic performance compared to other biomarkers [7,13]. In a recent report, we found overexpression of PIVKA-II in PDAC tissue and reduced circulating PIVKA-II levels after surgery in PDAC patients. The decrease in circulating PIVKA-II levels post-surgery suggests a reduction in tumor load. It can be speculated that baseline high PIVKA-II levels are a result of direct production by PDAC cells [14].

Considering these observations and the limited information available on the potential mechanisms underlying PIVKA-II’s diagnostic efficiency as a biomarker for PDAC, we transitioned from clinical observations to laboratory investigations to functionally characterize the role of this protein in this specific cancer type. Thus, in this study, our primary aim was to identify an in vitro model using stabilized PDAC cell lines to comprehensively characterize PIVKA-II expression, distribution, and the molecular mechanisms leading to its release. Within our experimental model, we also explored the relationship between PIVKA-II and glucose. Recent findings highlighted that 80% of PDAC patients exhibit glucose intolerance or frank diabetes, and the variability of glucose levels in PDAC cell lines has been associated with both tumor proliferation and metastasis [15,16,17]. Finally, considering the frequently attributed association between hyperglycemia and poor prognosis in PDAC due to glucose-dependent alterations in the epithelial–mesenchymal transition process (EMT) [18], we aimed to analyze the kinetics of PIVKA-II expression and release in correlation with EMT.

## 2. Results

### 2.1. PIVKA-II Expression in PDAC Cell Lines

In order to study the potential role of PIVKA-II as an early biomarker of pancreatic cancer in vitro, we examined the expression of PIVKA-II protein in a panel of PDAC cell lines whose main features are summarized in Table 1.

Cultured cells were lysed, separated on SDS-PAGE, and finally analyzed by Western blotting using Moab-anti PIVKA-II and anti b-actin. Considering that there are no cell lines known in the literature to be used as a PIVKA-II positive control, we separated a high-PIVKA-II titer serum from a PDAC patient previously determined by immunometry (CLEIA) using SDS-PAGE. Nontumorigenic HaCaT cells were used as a negative control because they do not express PIVKA-II. As shown in Figure 1A, pancreatic cell line PaCa 44, PANC-1, H-Paf II, and Capan-1 express PIVKA-II, although production levels are different for each cell type, as reported by the quantitative analysis of PIVKA-II protein calculated in relation to the b-actin detected (Figure 1B). Based on this result, we elected PANC-1 cells as a reference model to study PIVKA-II protein.

### 2.2. PIVKA-II Localization in PANC-1 Cells

In order to better characterize the cellular distribution of this novel biomarker, we analyzed PIVKA II localization in PANC-1 cells by using indirect immunofluorescence (IFA) [24]. Figure 2A shows an IFA performed on PANC-1 cells (left panels) and on HaCaT cells (negative control, right panels) labeled with the monoclonal antibody (mo-ab) directed against the PIVKA-II protein (red); nuclei were counterstained with DAPI (blue).

As shown in Figure 2, PIVKA-II expression seemed to be exclusive to PANC-1 cells and undetectable in HaCaT cells, thus confirming the immunoblotting results. PIVKA-II is mainly distributed in the cell cytoplasm, with an enrichment in the perinuclear zone of certain cells, and the distribution is granular and appears to be associated with the fibrous structures of the cells. This distribution is comparable to that observed in vivo [14].

To evaluate the organization of the actin cytoskeleton in PANC-1 cells, a morphological analysis was performed by double immunofluorescence microscopy. To this end, human PANC-1 cells and HaCaT cells were stained with PIVKA II (red) and phalloidin (green), which specifically recognize the filamentous actin cytoskeleton (Figure 2B). The morphological analysis shows no co-localization between actin and PIVKA II protein.

### 2.3. PIVKA-II Release in PANC-1 Cell Lines Is Glucose-Dependent

PIVKA-II is a free-circulating biomarker in vivo; however, nothing is yet known regarding the release mechanism of this protein. Several lines of evidence suggest that a large number (up to 80%) of pancreatic cancer patients suffer from hyperglycemia or diabetes, both characterized by elevated blood glucose levels [25]. Based on these observations, we wanted to investigate whether glucose could play a role in inducing the release of the PIVKA-II protein in vitro. To this end, we incubated PANC-1 cells in presence of increasing glucose doses (0 mM, 5 mM, 25 mM and 50 mM) for 24 h and 48 h. Following treatment, cells (Pell) and supernatants (Sup) were collected, separated by SDS-PAGE, and immunoblotted with different antibodies. As shown by the Western blotting analysis (Figure 3, left panel), we observed that PIVKA-II protein is released in the presence of 25 mM glucose following 48 h treatment, with an increase in the presence of 50 mM glucose; otherwise, this protein is retained in the absence of glucose or in the presence of very low concentrations of this sugar (5 mM) [16]. No increments in PIVKA-II production and release were observed following 24 h treatment. Ponceau staining of the immunoblot (red, lower left panel) was used as a supernatant loading control. In the same set of experiments, we also examined PANC-1 intracellular production of PIVKA-II protein in the presence of increasing glucose doses. Western blotting analysis of Figure 3B shows that the production of PIVKA-II seems to be directly related to the increase in glucose concentration in the culture medium, as also reported in the densitometric analysis presented in Figure 3C; b-actin was used as a loading control and human serum S191 was considered the positive control in the experiment. Taken together, these observations suggest that biomarker production and release in PANC-1 cells is promoted by glucose in a dose-dependent manner.

### 2.4. PIVKA-II Release in PANC-1 Cells Is Simultaneous with Epithelial–Mesenchymal Transition Activation

In order to study PIVKA-II release as a function of EMT onset, we evaluated the expression of two well-acknowledged EMT markers, Vimentin and Snail, in PANC-1 cells and in the presence of increasing doses of glucose [26]. As shown in Figure 3B, we observed the appearance of both Vimentin and Snail proteins in presence of 25 mM and 50 mM glucose. It is noteworthy that in this experiment, at the same glucose concentrations (i.e., 25 mM and 50 mM), PIVKA-II protein was released into the supernatant, thus suggesting that the biomarker in this in vitro model was released as EMT began; notably, PIVKA-II production in the PANC-1 cell line occurred before the complete activation of EMT.

## 3. Discussion

In developed nations, PDAC is presently ranked the fourth among the leading causes of cancer deaths; its mortality rate relative to its incidence has been a constant over the last two decades. Thus, improvements in timely PDAC diagnosis are mainly dictated by the necessity of setting up the decision-making process within a short space of time [27,28]. PDAC is characterized by an early and aggressive local invasion which, associated with a delayed clinical presentation and high metastatic potential, makes it a tumor with a poor prognosis [29]. In this scenario, circulating biomarkers, due to their availability, represent a powerful tool for early-stage diagnosis, prognosis, and follow up. Although many serum markers for the diagnosis of PDAC have been evaluated, to date, no reliable biomolecules (such as CEA, CA19.9, CA242) [30] have been identified for optimal clinical management. A notable push in this direction has been provided by recent studies focused on new biomolecules showing more reliable diagnostic performance [6,31,32]. Among these, the most promising is PIVKA-II protein, a modified prothrombin whose expression is related to vitamin K deficiency.

Recently, increasing attention has been paid to the relationship between vitamin K and malignancy [33]; several in vivo observational studies have established a relationship between vitamin K intake and cancer mortality [33,34]. An in vitro study also reported that vitamin K retains a peculiar cytotoxicity towards cancer cells through different mechanisms implicated in cell growth arrest and suppression of proliferation [33]. Taking these observations into account, all the products developed following vitamin K absence, such as PIVKA-II, acquire a new role in the management of cancer. This biomarker is commonly used for HCC diagnosis and prognosis, and recently altered values of PIVKA-II have been detected in various gastroenteric neoplasms such as gastric cancer, colon cancer, and PDAC, providing a new perspective on the diagnosis of these neoplasms [7,35]. The presence of this protein in PDAC is perhaps due to the fact that pancreas and liver tissues share a common embryological origin from the mesoderm, retaining a latent ability to trans-differentiate one into the other; it therefore seems reasonable to hypothesize that the characteristic expression in HCC could also be present in PDAC [36,37].

As a consequence of previous in vivo studies demonstrating that circulating levels of PIVKA-II were altered in patients with PDAC, here, we aimed to study the molecular aspect of PIVKA-II protein in vitro [14]. The in vitro study provided valuable information on the biological aspects of PIVKA-II; for the first time, in fact, we demonstrated that this protein is expressed in several cell lines originating from PDACs. Given the high basal level of protein expression, we chose PANC-1 cells as an in vitro model to study the properties of PIVKA-II in relation to PDAC. The peculiar cytoplasmic and granular distribution of PIVKA-II observed in vivo can be also confirmed in a PANC-1 model, thus strengthening previous reports’ findings and further supporting the choice of this cellular model. However, morphological analysis of PIVKA-II distribution has currently only provided us with a preliminary dataset, and further studies will be needed to clarify how the cellular structure is altered in response to an appropriate stimulus to facilitate the intracellular transport and release of the protein [24,38].

Currently, one of the hallmarks of PDAC is the cellular metabolism reprogramming promoted by mutations in the KRAS oncogene [39,40]. It is well documented that cancer cells use a large amount of glucose, which is processed to produce lactate even in the presence of oxygen, a process described as the Warburg effect [41]. Cancer cells are known to have markedly increased glycolytic flux even in the presence of oxygen and normal mitochondrial function. The main role of glycolytic flux in carbon metabolism is not limited to the adenosine triphosphate (ATP) production but is also pivotal for providing biomass for the anabolic processes that support cell proliferation. In 2011, Han et al. reported that proliferation of pancreatic cell lines was affected by different concentrations of glucose in a concentration-dependent manner [42]. Thus, in light of these observations, we investigated the effect of glucose on the PIVKA-II protein in our model. In our study, we applied the same experimental conditions of Han et al. [42] and we observed that increasing doses of glucose promote PIVKA-II cellular production in a dose-dependent manner. Similar experiments have also been conducted on PaCa44 cell lines but unfortunately, we did not observe any effects on PIVKA-II production or release. We can speculate that one possible explanation is related to the fact that while PANC-1 cells are responsive to glucose treatment [42], the other cell lines could activate PIVKA-II production and release through different stimuli.

Since there is no information in the literature on whether PIVKA-II could be considered an early or late disease marker, we studied its expression in relation to two established EMT-related molecules in order to evaluate if PIVKA-II release could be associated with EMT-dictated tumor progression. It has been demonstrated that high glucose levels could promote pancreatic cancer proliferation and invasion as well as EMT and metastasis [18]. EMT, the hallmark of cancer metastasis, is a complex developmental program in which epithelial cells lose many of their characteristics and acquire a mesenchymal phenotype that permits the invasion of surrounding tissues, distant metastasis, metabolic reprogramming, resistance to chemotherapy, and immune system suppression [26,43,44]. EMT is generally associated with a poor prognosis since the activation of this mechanism confers characteristic aggressiveness to the tumor [45,46], and in pancreatic cancer progression, it has been demonstrated that tumor seeding of distant organs occurs before and concomitantly with tumor formation at the primary site [47]. Biomarkers are widely used in EMT studies to characterize the state in which cells are found, and some of them are already associated with this process, such as growth factors (TFG-β and Wnts), transcription factors (SNAIL and TWIST), adhesion molecules (cadherins), and molecules present in the cytoskeleton (Vimentin) [48,49]. In our study, using the EMT biomarkers Snail and Vimentin, we observed that while PIVKA-II production take place independently of EMT onset, the release of this protein occurs concomitantly with the beginning of glucose-induced EMT. Taken together, these experimental data suggest that PIVKA-II may represent an early signal of cancer progression and can be considered a novel valuable tool for timely PDAC diagnosis. The overall information deriving from our study underlines the importance, in the field of biomedical research, of identifying preclinical experimental models that are useful both for characterizing specific cellular mechanisms involved in the progression of PDAC and for evaluating the effects of possible targeted therapeutic strategies. It must also be considered that in recent years, tumor markers have not only been used as disease indicators but are often molecules that actively participate in tumor progression [50].

This study does have some limitations. The primary limitation of this study is the use of a single cell line and the omission of investigating other models of carcinogenesis activation, such as a cytokine cocktail and hypoxia. Moreover, it will be important to make a comparison with results from other studies in this particular field. Notwithstanding, the results obtained in this research are to be considered preliminary; they offer promising new perspectives for establishing a new effective PDAC biomarker for early diagnosis.

Appropriate biomarkers are crucial for the screening, early diagnosis, treatment, and prognosis of PDAC. We believe that PIVKA-II could be a useful tool for PDAC screening in selected populations; particularly, it could be a valid aid for people at increased risk of developing PDAC, i.e., patients with diabetes or glucose intolerance.

The overall information derived from our study emphasizes the importance, in the field of biomedical research, of identifying preclinical experimental models that are useful both for characterizing specific cellular mechanisms involved in the progression of PDAC and for evaluating the effects of possible targeted therapeutic strategies.

## 4. Materials and Methods

### 4.1. Cell Culture and Treatments

The human PDAC-derived cell lines PANC-1 [19], PaCa44 [20], HPAF II [22] and Capan-1 [21] were cultured in RPMI 1640, 10% fetal calf serum (FCS) (Aurogene, Rome, Italy), Lglutamine (2 mM), streptomycin (100 mg/mL), and penicillin (100 U/mL) in 5% CO_2_ at 37 °C. Human keratinocytes (HaCaTs) are a spontaneously immortalized nontumorigenic human keratinocyte line [23] and were maintained in D-Mem, 10% heat-inactivated fetal bovine serum (Aurogene), 2 mg of L-glutammin (Aurogene), and penicillin/streptomycin (100 unit of penicillin, 100 mg/mL streptomycin) before being incubated at 37 °C in and 5% CO_2_. To study the effect of glucose concentration, cells were grown at 70% confluence on six wells washed with PBS and starved for 5 h at 37 °C in and 5% CO_2_. Following the starvation, complete medium (RPMI) was replaced in the presence of glucose concentrations varying from 5.0 to 50 mM for 12 h, 24 h, or 48 h.

### 4.2. Indirect Immunofluorescence (IFA)

Untreated or treated cells were seeded on sterilized coverslips and grown at 70% confluence than washed (PBS 1x), air-dried, fixed, and permeabilized as described elsewhere [24]. The following primary antibodies were used: mouse monoclonal anti-PIVKA-II (Biorbyt-Durham, NC, USA, 1:1000) and FITC-phalloidin (Sigma-Aldrich, St. Louis, MO, USA) (1:50). Sheep anti-mouse IgG-Cy3 (SAM-Cy3, Jackson-Ely, UK; 1:2000) antibodies were used as secondary antibodies. Nuclei were stained with DAPI (1:5000 in PBS, Sigma) 1 min RT. Immunofluorescence was analyzed by using an Axio Observer Z1 inverted microscope equipped with an ApoTome.2 System (Carl Zeiss Inc., Ober Kochen, Germany). Digital images were acquired with an AxioCam MRm high-resolution digital camera (Zeiss) and processed with the AxioVision 4.8.2 software (Zeiss). ApoTome optical sectioning images of fluorescent cells were recorded under 40/0.75 objective (Zeiss).

### 4.3. Western-Blot Analysis

Treated or untreated cells were washed in PBS 1X and lysed in a RIPA buffer 1x (150 mM NaCl, 1% NP-40, 50 mM TrisHCl, pH 8, 0.5% deoxycholic acid, 0.1% SDS, protease and phosphatase inhibitors) on ice for 30 min, as described elsewhere [24]. Protein concentration was measured by using a BCA protein assay kit (Sigma 71285-M) and 15 µg of protein was subjected to electrophoresis on 10% TGX FastCast (TGX FastCast Acrylamide Kits-Bio Rad, San Francicsco, CA, USA), according to the manufacturer’s instruction. The gels were transferred to nitrocellulose membranes (Bio-Rad, Hercules, CA, USA) for 45 min in Tris-glycine buffer, and the membranes were incubated in blocking solution (1 × PBS, 0.1% Tween20 and 3% of BSA (SERVA Electrophoresis GmbH, Heidelberg, Germany) containing the specific antibodies and developed using ECL Blotting Substrate (Advansta, San Jose, CA, USA). Concerning supernatants’ analysis, 2 mL out of 20 mL cell culture medium was loaded and separated by SDS-PAGE 10% (TGX FastCast-Kits-Bio Rad, San Francicsco, CA, USA) then immunoblotted on nitrocellulose membranes, as described elsewhere [24]. Membranes were then probed with anti PIVKA-II (Biorbyt 1:1000), anti-β-actin (Santa Cruz 1:1000, Santa Cruz, CA, USA), anti-Snail (Cell Signaling 1:100, Beverly, MA, USA), anti-Vimentin (Santa Cruz 1:200), polyclonal anti-mouse IgG-HRP (Bethyl, 1:10,000, Montgomery, TX, USA), and anti-rabbit IgG-HRP (Bethyl, 1:20,000). Detection was performed using Western Bright (Advansta, Menio Park, CA, USA)**.**

### 4.4. Densitometric Analysis

Quantification of protein bands was performed by densitometric analysis using Image J software (1.47 version, NIH, Bethesda, MD, USA), which was downloaded from the NIH website (http://imagej.nih.gov, accessed on 1 August 2022). The densitometric analysis was performed using GraphPad Prism 5.0 software (GraphPad Software Inc., La Jolla, CA, USA).

## Figures and Tables

**Figure 1 ijms-25-03498-f001:**
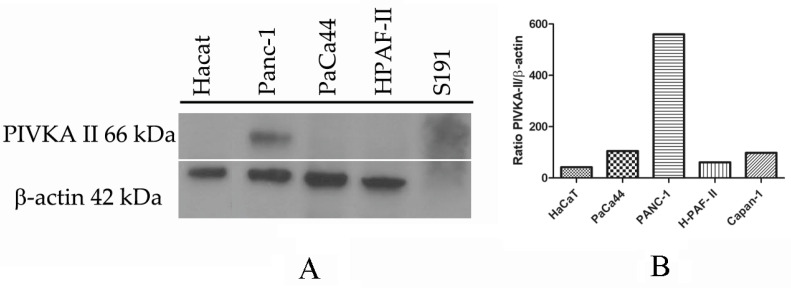
PIVKA-II protein expression in different cell lines. One representative experiment out of three is shown. (**A**) Western blot analysis of PIVKA-II protein in different pancreatic adenocarcinoma cell lines. (**B**) Densitometric evaluation of PIVKA-II protein in PDAC cell lines. Histograms represent the mean of the densitometric analysis of the ratio of PIVKA-II/β-actin. Densitometric analysis was performed with ImageJ software (1.47 version, NIH, Bethesda, MD, USA), which was downloaded from the NIH website (http://imagej.nih.gov, accessed on 1 August 2022) and plotted with GraphPad Prism 5.0 software.

**Figure 2 ijms-25-03498-f002:**
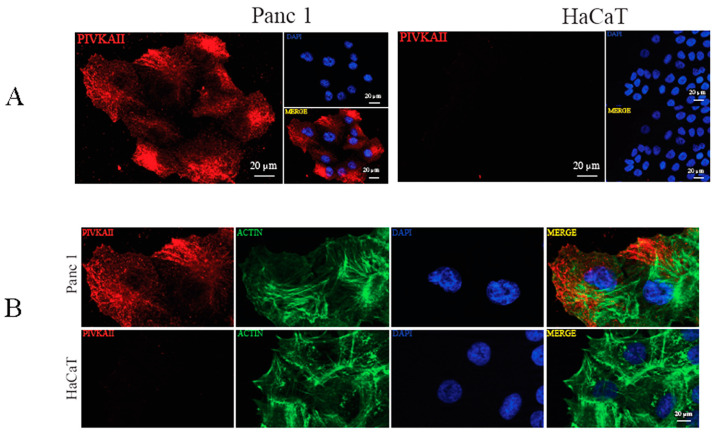
PIVKA-II localization in PANC-1 cells. (**A**) IFA performed on PANC-1 (left panel) and HaCaT (right panel) cell lines, showing PIVKA-II (red) and nuclei (blue). Representative images out of three are shown. (**B**) Double IFA performed on PANC-1 (upper panel) and HaCaT (lower panel) cell lines, showing PIVKA-II (red) and nuclei (blue) and actin (green). Representative images out of three are shown.

**Figure 3 ijms-25-03498-f003:**
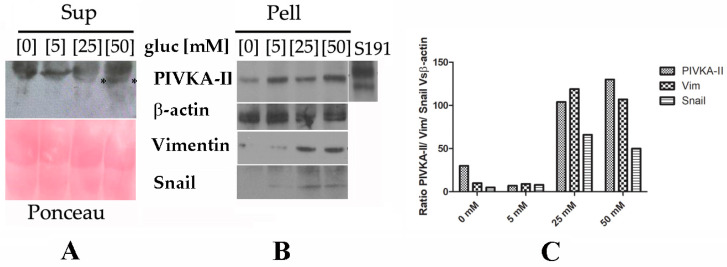
Expression and release of PIVKA-II in the presence of increasing doses of glucose in PANC-1 cells. One representative experiment out of three is shown. (**A**,**B**) Western blotting analysis performed on PANC-1 cells treated in the presence of increasing doses of glucose (5 mM, 25 mM, 50 mM) for 48 h. Following the treatment, Super (**A**) and Pellet (**B**) were recovered, separated on 12% SDS-PAGE, and analyzed by Western blotting with the indicated antibodies. Asterisks (*) in Figure 3A indicates released PIVKA-II protein. Correct loading of the supernatants was checked by Ponceau staining of the membrane (pink panel). b-actin was used as the lysates’ loading control. (**C**) Densitometric evaluation of PIVKA-II protein in different cell lines. Histograms represent the mean of the densitometric analysis (performed with ImageJ) of the ratio of PIVKA-II–Vimentin–Snail vs. β-actin.

**Table 1 ijms-25-03498-t001:** General characteristics of cell lines selected for this experimental study.

Cell Line	Age	Gender	Origin	Cell Type	Mutations
PANC-1 [19,20]	56	F	Primary tumor	Epithelial	KRAS, TP53, CDKN2A/p16
PaCa44 [21]	65	M	Primary tumor	Epithelial	KRAS, TP53, CDKN2A/p16
H-PAF-II [22]	44	M	Ascites	Epithelial	KRAS, TP53, CDKN2A/p16
Capan-1 [22]	40	M	Liver metastasis	Epithelial	KRAS, TP53, CDKN2A/p16 SMAD4/DPC4
HaCaT [23]	nd	nd	Keratynocyte	Epithelial	

## Data Availability

Data are contained within the article.

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
