# Peer review of "PANC-1 Cell Line as an Experimental Model for Characterizing PIVKA-II Production, Distribution, and Molecular Mechanisms Leading to Protein Release in PDAC"

_ijms, 2024, doi:10.3390/ijms25063498_

Round 1
Reviewer 1 Report
Comments and Suggestions for Authors
This study investigates Protein Induced by Vitamin K Absence-II (PIVKA-II) as a potential biomarker for pancreatic ductal adenocarcinoma (PDAC) through experiments on PANC-1 cell lines. The research explores PIVKA-II's synthesis, distribution, and the cellular processes that facilitate its secretion, particularly examining the influence of glucose on PIVKA-II levels and its association with epithelial-mesenchymal transition (EMT). The authors propose PIVKA-II as a candidate for early PDAC detection, emphasizing its glucose-mediated release and concurrent activation with EMT markers. Nevertheless, the manuscript's persuasive power is weakened by the absence of robust experimental support and the substandard quality of the available data, rendering it unsuitable for publication in International Journal of Molecular Sciences.
1. The manuscript asserts significant PIVKA-II expression in the Panc-1 cell line as depicted in Figure 1, yet the actin control is noticeably faint and unclear, particularly in the full gel presented in the supplementary information. This raises concerns about the appropriateness of the actin antibody used. It is crucial for the authors to verify the quality of their control protein antibody and adjust protein concentrations to achieve distinct control band visibility. This recommendation applies to all blots throughout the document.
2. The identification of the PIVKA-II band is questionable due to the unclear nature of the blot. The issue does not appear to be antibody-related, as Figure 3b shows a single major PIVKA-II band. Authors must provide convincing evidence that the band they are targeting is indeed PIVKA-II and not just one of many indistinct bands.
3. The paper suggests that PIVKA-II expression in Panc-1 cells is granular and linked to the cells' fibrous structures. To substantiate this claim, the authors are encouraged to include markers for fibrous structures, such as phalloidin, in their analysis.
4. The manuscript alleges to have conducted a kinetic study on PIVKA-II release, yet fails to present essential kinetic parameters like pKa, Km, or Vmax. Without these values, it is inaccurate to describe the work as a kinetic study.
5. Figures 3a and 3b show no discernible difference in PIVKA-II band intensity across glucose treatments ranging from 0 to 50 mM and 5 to 50 mM, respectively. Despite this, the authors conclude that PIVKA-II expression increases with glucose treatment. This claim is inconsistent with the presented data.
6. All bar plots should be created using professional graphing software. Photographs of bar plots are not acceptable for publication.
Comments on the Quality of English LanguageThe English language usage in the manuscript requires significant improvement for clarity and professionalism.
Author Response
Rew 1
This study investigates Protein Induced by Vitamin K Absence-II (PIVKA-II) as a potential biomarker for pancreatic ductal adenocarcinoma (PDAC) through experiments on PANC-1 cell lines. The research explores PIVKA-II's synthesis, distribution, and the cellular processes that facilitate its secretion, particularly examining the influence of glucose on PIVKA-II levels and its association with epithelial-mesenchymal transition (EMT). The authors propose PIVKA-II as a candidate for early PDAC detection, emphasizing its glucose-mediated release and concurrent activation with EMT markers. Nevertheless, the manuscript's persuasive power is weakened by the absence of robust experimental support and the substandard quality of the available data, rendering it unsuitable for publication in International Journal of Molecular Sciences.
- The manuscript asserts significant PIVKA-II expression in the Panc-1 cell line as depicted in Figure 1, yet the actin control is noticeably faint and unclear, particularly in the full gel presented in the supplementary information. This raises concerns about the appropriateness of the actin antibody used. It is crucial for the authors to verify the quality of their control protein antibody and adjust protein concentrations to achieve distinct control band visibility. This recommendation applies to all blots throughout the document.
Thank you for your observation, according to your suggestion Fig 1 has been changed.
- The identification of the PIVKA-II band is questionable due to the unclear nature of the blot. The issue does not appear to be antibody-related, as Figure 3b shows a single major PIVKA-II band. Authors must provide convincing evidence that the band they are targeting is indeed PIVKA-II and not just one of many indistinct bands.
Thank you for your comment, probably the illustration of our Western Blot wasn’t clear enough. In our experiment we aimed to investigate the effect of a gradual glucose increase in PANC-1 cells. After treating cells with increasing glucose doses, they were separated from the supernatant and analyzed by Western Blot. PANC-1 cell lysates were separated respectively by SDS-PAGE on a 12 % polyacrylamide gel optimized to yield the best separation of Snail (29 kDa) and by SDS-PAGE on a 10% polyacrylamide gel optimized to yield the best separation of PIVKA-II, Vimentin and b-actin (respectively 66kDa, 57 kDa and 42 kDa). Conversely, as shown in left panel of the figure (3a), we separated by SDS-PAGE cell supernatant (20ul out of 2ml). As reported in literature, the quantitative analysis of proteins secreted from the cells poses a challenge due to their low abundance and the interfering presence of a large amount of bovine serum albumin (BSA 67kDa) in the cell culture media ( Trajkovic K, Jeong H, Krainc D. Mutant Huntingtin Secretion in Neuro2A Cells and Rat Primary Cortical Neurons. Bio Protoc. 2018 Jan 5;8(1):e2675. doi: 10.21769/BioProtoc.2675. PMID: 29326963; PMCID: PMC5759345.) In our specific case, this problem is enhanced by the fact that PIVKA-II is 66 kDa and albumin is 67 kDa, so they are extremely complicated to differentiate. However, we used an highly specific PIVKA-II antibody, as you can see in cells lysates. In the western blot we are showing a 67kDa band which is predominant and present in all sample with the same intensity. This band is particularly prominent also in Ponceau. Despite the presence of this aspecific protein, in supernatants we still describe a 66kDa band (asterisk in figure), specifically recognized by anti PIVKA-II antibody, exclusively present in the sample treated with 25mM and 50 mM of glucose, while it is not shown in time 0 samples or in samples treated with only 5 mM of glucose. So, we agree that the presence of an aspecific band is an intrinsic limit of used method but we believe that its presence isn’t a factor modifying our results.
- The paper suggests that PIVKA-II expression in Panc-1 cells is granular and linked to the cells' fibrous structures. To substantiate this claim, the authors are encouraged to include markers for fibrous structures, such as phalloidin, in their analysis.
Thank you for your important suggestion. We added as you requested in the manuscript and Fig 2 has been completed with phalloidin staining as detailed in figure legend.
- The manuscript alleges to have conducted a kinetic study on PIVKA-II release, yet fails to present essential kinetic parameters like pKa, Km, or Vmax. Without these values, it is inaccurate to describe the work as a kinetic study.
Thank you for your interesting observation. We agree that it would be inappropriate to describe the work as a kinetic study so, following your suggestion, we changed the sentence in the manuscript describing the test as a glucose-dose-dependent one
- Figures 3a and 3b show no discernible difference in PIVKA-II band intensity across glucose treatments ranging from 0 to 50 mM and 5 to 50 mM, respectively. Despite this, the authors conclude that PIVKA-II expression increases with glucose treatment. This claim is inconsistent with the presented data.
Thank you for your comment. We are aware that with this type of blot it could be challenging to discern differences between bands. Nevertheless, to solve this problem, to facilitate the discrimination of PIVKA-II band intensity we also used densitometry. The measurement of housekeeping proteins by densitometry is commonly used for Western Blot normalization, in our case we used b-actin. Also, as reported in densitometric analisys, PIVKA-II, Vimentin and Snail bands were respectively quantified in each lane. This quantization allowed to highlight that PIVKA-II increase is concomitant to the addition of raising glucose concentrations. Additionally, it’s important to mention that our results were obtained in 3 different independent experiments.
- All bar plots should be created using professional graphing software. Photographs of bar plots are not acceptable for publication.
Thank you for your suggestion, we have corrected densitometric plots by using GraphPad Prism 5.0 software.
Comments on the Quality of English Language
The English language usage in the manuscript requires significant improvement for clarity and professionalism.
The English has been revised by a native speaking Professor.
Reviewer 2 Report
Comments and Suggestions for Authors
Farina A et al. showed an interesting new biomarker, protein induced by vitamin K absence II (PIVKA-II) role in pancreatic cancer(PC). The protein sensitivity and specificity were 78.67 % and 90.67% respectivly reported in pancreatic cancer(iLIVER Volume 1, Issue 4, December 2022, Pages 209-216). However, the authors need to address few minor points
1. Abstract: Please address issues happen could not find any results?? Need to revise structured format???
2. The authors need to screen more pancreatic cancer cells BxPC3; CD18/HPAF;MIA PaCa-2; SUIT2;SW1990 to check the PIVKA-II expression ??
3. The study need more data to address why did bring this secreated biomarker to study human PC cells?
Author Response
REW 2
Farina A et al. showed an interesting new biomarker, protein induced by vitamin K absence II (PIVKA-II) role in pancreatic cancer(PC). The protein sensitivity and specificity were 78.67 % and 90.67% respectivly reported in pancreatic cancer (iLIVER Volume 1, Issue 4, December 2022, Pages 209-216). However, the authors need to address few minor points
- Abstract: Please address issues happen could not find any results?? Need to revise structured format???
Thank you for your observation. We apologize for the poorly constructed abstract , it has been revised and changed.
The authors need to screen more pancreatic cancer cells BxPC3; CD18/HPAF;MIA PaCa-2; SUIT2;SW1990 to check the PIVKA-II expression ??
Thank you for your question, this is a very interesting point that we can consider as a limit of this work. As outlined in the text, we analyzed additional pancreatic cell lines (Fig 1) that unfortunately exhibited low levels of PIVKAII expression. Furthermore, we investigated both the expression and release of PIVKA-II in the cell supernatant of PANC-1 and PaCa44. It is noteworthy that in the case of PaCa44, PIVKA-II production and release remained unaffected by glucose treatment. Ongoing experiments are being conducted to provide a more comprehensive understanding of this phenomenon on several cell lines.
The study need more data to address why did bring this secreated biomarker to study human PC cells?
Thank you for your comment. In this manuscript, we are pioneering in bringing attention to the existence of this protein in cell lines derived from pancreatic tumors. Our primary objective has been to evaluate and characterize the presence of PIVKA-II specifically in PDAC cell lines. Our study serves as an initial step towards a deeper comprehension of the role of PIVKA-II in PDAC and the significance associated with its expression. In the context of this study, which aims to investigate biomarkers, PIVKA-II serves as a biomolecule released as a consequence of a mechanism associated with carcinogenesis.
Round 2
Reviewer 1 Report
Comments and Suggestions for Authors
The authors have adequately addressed the comments and concerns raised by the reviewers, making the manuscript now suitable for publication in International Journal of Molecular Sciences.
Comments on the Quality of English LanguageThe manuscript's English description is clear and professional.